# Evaluating the Efficacy of Eccentric Half-Squats for Post-Activation Performance Enhancement in Jump Ability in Male Jumpers

Theodoros M. Kannas [1], Georgios Chalatzoglidis [1], Elli Arvanitidou [1], Nicolas Babault [2,3], Christos Paizis [2,3,*] and Fotini Arabatzi [1]

[1]  Laboratory of Neuromechanics, Department of Physical Education and Sport Science, Aristotle University of Thessaloniki, Agios Ioannis, GR-62122 Serres, Greece; thkannas@phed-sr.auth.gr (T.M.K.); gchalatzo@phed-sr.auth.gr (G.C.); arvan7elli@yahoo.gr (E.A.); farabaji@phed-sr.auth.gr (F.A.)
[2]  INSERM UMR1093-CAPS, UFR des Sciences du Sport, Université de Bourgogne, F-21000 Dijon, France; nicolas.babault@u-bourgogne.fr
[3]  Centre d'Expertise de la Performance, UFR des Sciences du Sport, Université de Bourgogne, F-21000 Dijon, France
*  Correspondence: christos.paizis@u-bourgogne.fr; Tel.: +33-3-80-39-67-29

**Abstract:** The purpose of the present study was to investigate effect of post-activation performance enhancement (PAPE) induced by the eccentric half-squat exercise on vertical jump performance in male jumpers. The jumping height, peak power, and work were measured and evaluated in twenty male jumpers (age: $21.2 \pm 1.7$ years, height: $191.1 \pm 3.3$ cm, body mass: $81.56 \pm 7.3$ kg) who participated in the national championship last year. Participants performed five eccentric half-squats at 85% of their one-repetition maximum (1 RM), with a knee angle below $90°$, followed by immediate and 2 min delayed jump assessments using the Squat Jump (SJ) and Countermovement Jump (CMJ) tests. Results showed that this specific PAPE protocol did not significantly improve jump performance for the SJ (Height: ES = 0.613, $p = 0.462$, Work: ES = 0.124, $p = 0.231$, Power: ES = 0.382, $p = 0.125$) or CMJ (Height: ES = 0.523, $p = 0.368$, Work ecc: ES = 0.133, $p = 0.505$ (only main effect time $p < 0.05$), Work con: ES = 0.114, $p = 0.101$, Power ecc: ES = 0.134, $p = 0.177$, Power con: ES = 0.182, $p = 0.195$, Leg stiffness: ES = 0.095, $p = 0.358$) tests. Factors such as stimulus specificity, rest intervals, muscle length, and the balance between potentiation and fatigue may explain these results. This study highlights the complexity of PAPE responses and suggests that a single set of eccentric squats with a short rest may not improve jump performance in male jumpers. Further research is required to optimize the interplay between conditioning stimuli and rest periods to maximize PAPE effects in athletic performance enhancement.

**Keywords:** post-activation performance enhancement; eccentric half-squat; squat jump; countermovement jump; jumping height; rest

## 1. Introduction

Several training methods are effective in increasing muscular power and improving athletic performance. Post-activation potentiation enhancement (PAPE) has been suggested as a means to acutely enhance short-duration athletic performance that relies on maximal power production. It refers to the acute improvement in muscular function based on its contractile history, strongly influenced by factors including muscle temperature changes, intramuscular fluid accumulation, and muscle activation [1,2]. The conditioning stimulus to induce PAPE can be achieved through different methods and modes of activities, including isometric exercises, medium or heavy resistance conditioning, and plyometrics, leading to a higher rate of force development or greater jumping height [3,4] and sprint performance [5,6]. Male elite rugby players experienced a 5% increase in jumping height [7],

and male adolescent soccer players showed a significant improvement in repeated sprint performance after performing loaded back squats [8]. Even in young female soccer players, combined balance and strength training acted as an effective PAPE stimulus to enhance jumping performance [9]. These findings highlight the potential impact of PAPE on sports activities with high power demands.

The importance of power and work production, coupled with the levels of leg stiffness during an athletic performance, has been previously described [10–14]. Power production in eccentric and concentric phases has been described as a determined factor in the final jumping performance [15,16]. The ability to rapidly produce high levels of force is directly linked to the capacity for work, allowing athletes to execute powerful jumps. The optimal level of leg stiffness ensures the proper transmission of the body from the eccentric to the concentric phase of the stretch-shortening cycle, leading to an efficient transfer of the produced work [14,17]. However, despite the importance of these parameters to the jumping performance, the final jumping height seems to be the best indicator of an optimal use of the produced force and an efficient transmission of produced power and work. Given the above, the analysis of the kinetic parameters provides important insights into the jumping performance; however, the final jumping height seems to be the best indicator of an optimal jump.

Isometric exercises have been effective in triggering PAPE responses, especially in individuals with high-strength capabilities [1]. A previous study suggested that 15 short, intermittent, and repetitive maximal voluntary contractions (MVCs) provide the most effective isometric stimulus to induce PAPE [18]. A $3 \times 3$ s stimulus of an isometric half-squat led to increased jumping performance due to a greater rate of force development in trained individuals [1]. Additionally, plyometric exercises effectively produce acute improvements in countermovement jump and sprint performance [19]. They may even offer similar benefits compared to high-loaded resistance exercises when used as conditioning activities by recreational athletes before running [20]. Determining the best interaction between the conditioning stimulus and rest period is a determined factor in PAPE responses [21]. This interaction depends on several factors, such as the type and intensity of the stimulus and the training background of the participants.

The high-loaded resistance stimulus typically involves multi-joint free-weight exercises with loads exceeding 85% RM (Repetition Maximum), which has gained widespread popularity as a potential effective stimulus, to enhance strength and power performance due to PAPE [22–24]. Such loading of the muscle–tendon system is known to enhance subsequent higher-velocity exercises like sprints or vertical jumps [25,26]. A previous study has shown that back squats enhanced jumping height and power production during countermovement jumps, particularly among international sprint swimmers [27]. Similarly, in rugby players, PAPE was observed after squats at nearly twice their body weight [7]. Additionally, significant increases in countermovement jump height were reported, attributed to greater muscular activity in the gluteus during parallel squats [28]. On the contrary, a recent study found that eccentric and concentric squats did not impact final countermovement jump performance in track and field athletes [29]. Despite the efficacy of high-loaded exercises in inducing the PAPE effect, their practical applicability remains uncertain. The data on the effects of strength exercises on subsequent jump performance, particularly the interaction between the type of stimulus and optimal rest periods to maximize power, are limited and warrant further investigation.

Trained athletes specializing in jumping events and sprinting often include plyometrics and heavy resistance training in their workout routines. These athletes usually show neural activation [30], architectural features [31], and a higher proportion of fast muscle fibers [32], all influencing their power production during athletic events. Chronic resistance training increases their resistance to fatigue through improved buffering capacity [23,33] and increased protection against skeletal muscle damage [34]. Consequently, it is reasonable to assume that conditioned athletes experience less fatigue than non-athletes, leading to the likelihood of the PAPE effect occurring closer to the conditioning stimulus. On the other

hand, athletes with predominantly fast-twitch fibers might be negatively impacted by short rest periods between the conditioning stimulus and their athletic performance due to the limited fatigability of these fibers. From a practical standpoint, rest periods of 2 to 5 min highly align with the common rest period provided to finalists during jumping events.

The research hypothesis of the present study was that eccentric conditioning activity could lead to an increase in the performance of jumping ability after a short rest period. Hence, the primary aim of this research was to assess the impact of PAPE induced by the half-squat exercise on the vertical jumping performance (including jump height, peak power, and work) in male jumpers, utilizing a brief rest interval.

## 2. Materials and Methods

### 2.1. Participants

A power analysis using G*Power 3.1.9.7 software guided our study's sample size determination. The analysis was based on a repeated-measures ANOVA design, with effect size: 0.35, number of conditions: 2 (intervention and control), number of measures: 3 (PRE, POST1, and POST2), correlation among repeated measures: 0.6 (Estimated by ICC). The power analysis recommended a sample size of 20 participants to achieve 97% statistical power. Twenty male athletes (age: $21.2 \pm 1.7$ years, height: $191.1 \pm 3.3$ cm, body mass: $81.56 \pm 7.3$ kg) with a minimum of 12 training hours per week, signed informed consent forms before their inclusion in the study. All the participants were athletes in jumping events in track and field (12 long jumpers and 8 high jumpers) who participated in the national championship last year. They had been engaged in regular strength training for at least 3 years. Furthermore, these athletes had undergone at least two lower extremity strength development training sessions during the period of peak strength increase in the last 2 years. Approval for the experiment was obtained from the University Ethics Committee on Human Research (ERC-008/2020). None of the participants were involved in any particular strength and/or plyometric training program four days before the experimental sessions.

### 2.2. Experimental Procedure

Participants visited the laboratory on five separate days (Figure 1). During the first visit, the maximal load for a single repetition (1 RM, MVC) was determined, starting with ten repetitions and progressively reducing to one [35,36]. The 1 RM was represented by the heaviest weight a participant could lift once using the correct lifting technique, without any additional movements. Additionally, they familiarized themselves with the testing setup (Squat Jump (SJ), Countermovement Jump (CMJ)) and the stimulus procedure with the eccentric half-squat exercise. All the participants performed a sub-maximal (ranging from 50% to 65% of the measured MVC) eccentric half-squat.

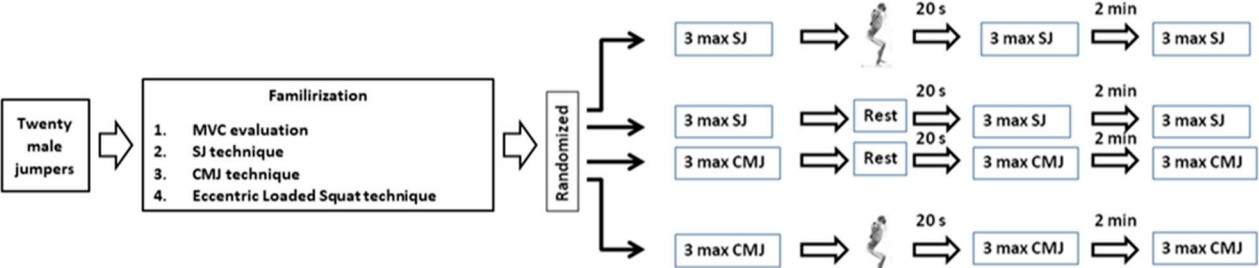

**Figure 1.** The whole study procedure (MVC: Maximum Voluntary Contraction—1 RM, SJ: Squat Jump, CMJ: Counter Movement Jump).

All jumps were performed on a three-dimensional platform (Kistler Type 9281C, Kistler Instruments, Winterthur, Switzerland) with ground reaction forces (GRF) to be recorded [37]. The subsequent four visits were designated as experimental sessions, with two sessions

serving as controls (involving no conditioning activity, equating to rest) and the other two as conditioning sessions (involving post-activation potentiation exercise—PAPE, specifically eccentric half-squat). The sequence of these sessions was randomized. Furthermore, Squat Jumps were assessed during two sessions (one control and one post-activation potentiation exercise), while Countermovement Jumps were evaluated during the other two.

All test days started with a 6 min warm-up routine on a static bicycle. Following the warm-up, the baseline of the jumping test (depending on the day) was performed (PRE) and repeated after the conditioning stimulus (POST1). Then, volunteers rested for 2 min (POST2), and they repeated the jumps. The best of three jumps, based on maximum height output, was used for further analysis. The maximum power, maximum work of the eccentric and concentric phases, and lower limb stiffness were analyzed and evaluated.

### 2.3. Conditioning Activity Procedure

The conditioning stimulus included 5 repetitions of eccentric half-squats at 85% of the MVC, with the knee joint angle reaching lower than 90° and each repetition lasting 4″ (almost 30°/s) and regulated by a metronome. The bar was placed on the stops of the squat rack at the end of the eccentric phase while the assistants lifted the Olympic bar to the initial position. During jump tests, the participants performed three SJs or CMJs with 15 s intervals between the jumps (Figure 1). The total SJ or CMJ test duration, including the rest intervals and duration of the jumps, did not exceed 90 s each time. In the control session, the same procedure was followed, and the performance of the jumps was evaluated at the same time points, except that the participants rested passively (seated) during the conditioning activity. All the sessions were conducted at approximately the same time of day for all participants, and there was no control for diet or hydration (Figure 1).

### 2.4. Power, Work, and Leg Stiffness Calculation

The total vertical displacement of the subject's COM ($\Delta l$) during the ground contact phase was calculated by double integration of the vertical acceleration over time. Leg stiffness was calculated from the ratio of ground reaction force to $\Delta l$ at the instant that vertical velocity was equal to zero when the COM reached its lowest point from the standing position, and the leg spring was maximally compressed. The peak vertical ground reaction force ($F_{peak}$) was obtained from the force–time curve, usually while the leg was maximally compressed ($F_{peak}$ may have also been displayed either slightly before or immediately after the lower positions).

$$k_{leg} = F_{peak} / \Delta l \tag{1}$$

The acceleration was integrated to give the speed. From this, power could be calculated using the equation $P = \vec{F} \cdot \vec{v}$, where $F$ is the force measured by the force plate and $v$ is the velocity calculated above. Work was calculated using the following equation:

$$W = \int_{t_i}^{t_f} \left( \vec{F} \cdot \vec{v} \right) dt, \tag{2}$$

where $t$ was time.

### 2.5. Statistical Analysis

Data analysis was performed using SPSS (IBM Corp., Version 25.0, Armonk, NY, USA). The statistical approach included separate analyses for SJ and CMJ. For each jump type, a repeated-measures ANOVA ($2 \times 3$) was conducted. This analysis incorporated the ICC-calculated correlation among repeated measures, accommodating within-subject effects and potential interactions between time points and conditions. To assess the practical significance of the observed differences, effect sizes were calculated. Additionally, post hoc analyses were conducted to explore the nature of the significant interaction between conditions and time points. Notable differences between PRE, POST1, and POST2 jumps were found for key parameters, including maximum power, work, maximum jump height,

and lower limb stiffness. This comprehensive approach to data analysis adds depth to the interpretation of the results, helping to understand whether there are significant differences and the nature and magnitude of these differences. A significance level of $p < 0.05$ was chosen to assess statistical significance.

## 3. Results

### 3.1. Intraclass Correlation Coefficients

The Intraclass Correlation Coefficients (ICCs) for the Squat Jump (SJ) and Countermovement Jump (CMJ) measurements yielded values of 0.92 and 0.89, respectively, indicating high reliability and consistency in these assessment techniques.

### 3.2. Squat Jump

The ANOVA showed no statistically significant differences in the maximum SJ height with respect to time, the three measurements ($F(1,18) = 1.3$, $p = 0.458$; Figure 2), and the interaction between conditions and time ($F(1,18) = 3.2$, $p = 0.062$).

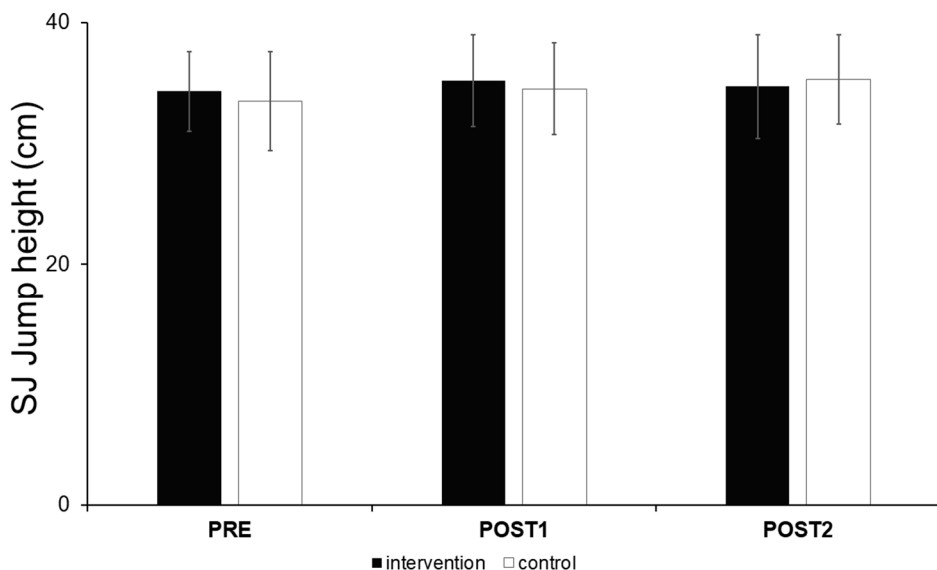

**Figure 2.** Jumping height (mean ± SD) during SJ (Squat Jump) at the three timepoints.

In addition, there was no statistically significant difference in the produced SJ-positive work in time ($F(2,18) = 0.3$, $p = 0.853$) and interaction ($F(2,18) = 1.4$, $p = 0.343$). Finally, there was no statistically significant difference in the maximum power between the PRE, POST1, and POST2 jumps (Figure 3) in either of the two conditions ($F(2,18) = 0.6$, $p = 0.526$) and the interaction between conditions and time ($F(2,18) = 0.6$, $p = 0.316$; Table 1).

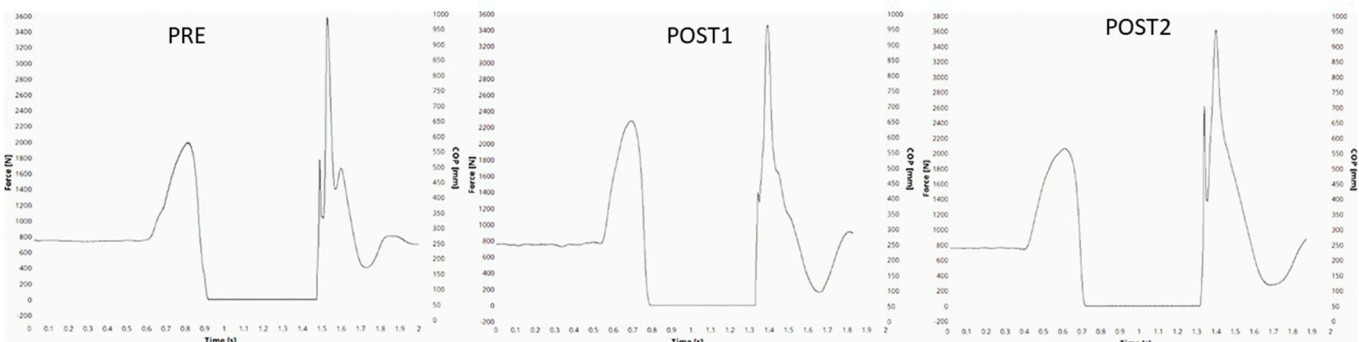

**Figure 3.** Representative (one subject) force−time curve of Squat Jump before (PRE), 20 s (POST1), and 2 min after the eccentric semi−squat (POST2).

**Table 1.** Squat Jump (SJ) performance for each condition, timepoint, and tests of within-subject contrasts.

|  |  | PRE | POST1 | POST2 | *p*-Values/ES (ηp²) Interaction |
|---|---|---|---|---|---|
| Jump Height (cm) | Intervention | 34.3 ± 3.3 | 35.2 ± 3.8 | 34.7 ± 4.3 | 0.462/0.613 |
|  | Control | 33.5 ± 4.1 | 34.5 ± 3.8 | 35.3 ± 3.7 |  |
| Work Con (J) | Intervention | 457.3 ± 58.8 | 463.2 ± 48.7 | 447.4 ± 52.5 | 0.231/0.124 |
|  | Control | 486.6 ± 35.8 | 478 ± 46.7 | 460.8 ± 43.5 |  |
| Power Con (w) | Intervention | 1589.9 ± 276.4 | 1602.5 ± 317.3 | 1595 ± 243.5 | 0.125/0.382 |
|  | Control | 1605.8 ± 246.4 | 1611 ± 367.6 | 1628.6 ± 213.3 |  |

### 3.3. Countermovement Jump

The ANOVA showed no statistically significant differences in the maximum CMJ height with respect to time and the three measurements (F(1,18) = 1.1, *p* = 0.368; Figure 4) and the interaction between groups and time (F(1,18) = 3.7, *p* = 0.053; Table 2).

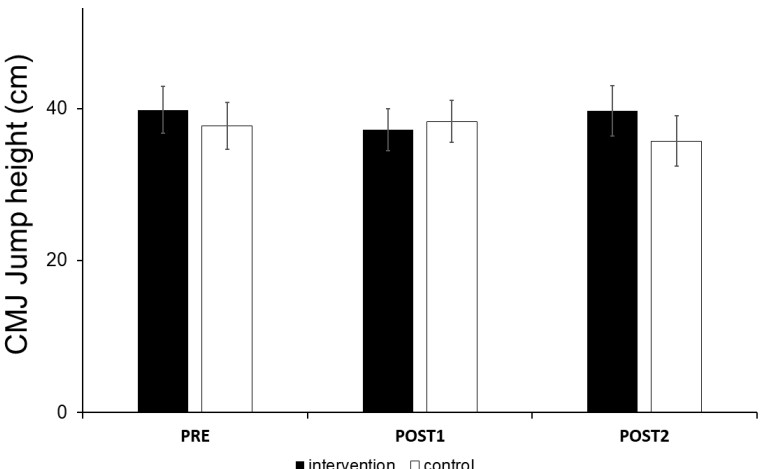

**Figure 4.** Jumping height (mean ± SD) during CMJ (Countermovement Jump) at the three timepoints.

**Table 2.** Countermovement Jump (CMJ) performance for each condition, time point, and tests of within-subject contrasts.

|  |  | PRE | POST1 | POST2 | *p*-Values/ES (ηp²) Interaction |
|---|---|---|---|---|---|
| Jump Height (cm) | Intervention | 39.8 ± 3.1 | 37.2 ± 2.8 | 39.7 ± 3.3 | 0.368/0.523 |
|  | Control | 37.7 ± 3.1 | 38.3 ± 2.8 | 35.7 ± 3.3 |  |
| Work Ecc (J) | Intervention | 93.9 ± 11 | 101 ± 11.9 | 110.5 ± 9.9 | 0.505/0.133 |
|  | Control | 85.4 ± 11 | 99.2 ± 11.9 | 99.2 ± 9.9 |  |
| Work Con (J) | Intervention | 347.7 ± 28.8 | 354.2 ± 28.7 | 340.4 ± 30.5 | 0.101/0.114 |
|  | Control | 376.6 ± 28.8 | 379 ± 28.7 | 385.8 ± 30.5 |  |
| Power Ecc (w) | Intervention | −2196 ± 109.9 | −2230.1 ± 104.1 | −2195.5 ± 110.7 | 0.177/0.134 |
|  | Control | −2175 ± 109.9 | −2225.2 ± 104.1 | −2255.6 ± 110.7 |  |
| Power Con (w) | Intervention | 2341.9 ± 176.4 | 2379.5 ± 167.6 | 2249 ± 173.3 | 0.195/0.182 |
|  | Control | 2510.8 ± 176.4 | 2538 ± 167.6 | 2549.6 ± 173.3 |  |
| Lower Leg Stiffness (N/cm) | Intervention | 26 ± 2.5 | 27.5 ± 2.6 | 27.5 ± 2.7 | 0.358/0.095 |
|  | Control | 25.7 ± 2.5 | 25.7 ± 2.6 | 26 ± 2.7 |  |

Values: means ± standard deviations. Ecc: eccentric phase. Con: concentric phase. ES: effect size.

The work during the eccentric phase revealed a significant main effect of time ($F_{(2,18)}$ = 9.9, $p$ = 0.002). Post hoc pairwise comparisons using Bonferroni revealed a significant mean difference between PRE (M = 89.7 J) and POST1 (M = 100.1 J; $p$ = 0.027) and between PRE and POST2 (M = 104.9; $p$ = 0.008), while no significant mean difference was found between POST1 and POST2 ($p$ = 0.562) for eccentric work (Figure 5). However, there was no interaction between group and time ($F_{(2,18)}$ = 0.7, $p$ = 0.505; Table 2). In contrast, there was no statistically significant difference in produced concentric work in time ($F_{(2,18)}$ = 0.4, $p$ = 0.708) and interaction ($F_{(2,18)}$ = 2.6, $p$ = 0.101; Table 2). Power was not statistically different during the eccentric and concentric phases across time (Table 2) and all three measures ($F_{(2,18)}$ = 1.6, $p$ = 0.236; $F_{(2,18)}$ = 1.9, $p$ = 0.177, respectively) and the interaction between group and time ($F_{(2,18)}$ = 1.6, $p$ = 0.237; $F_{(2,18)}$ = 1.8, $p$ = 0.195, respectively). Finally, the lower limb stiffness during the eccentric phase did not show any statistical differences with time ($F_{(2,18)}$ = 1.2, $p$ = 0.336) or the condition x time interaction ($F_{(2,18)}$ = 1.1, $p$ = 0.358; Table 2).

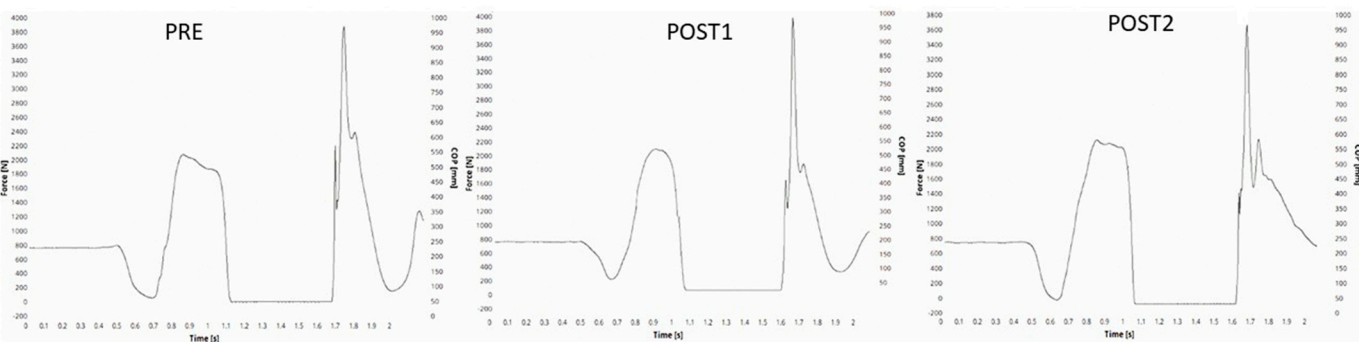

**Figure 5.** Representative (one subject) force−time curve of Countermovement Jump before (PRE), 20 s (POST1), and 2 min after the eccentric semi−squat (POST2).

## 4. Discussion

The objective of this study was to investigate whether the eccentric back-squat followed by 20 s and 2 min rest periods, as a preliminary conditioning stimulus, could improve vertical jumping performance. The findings of the present study indicated that the 2 min rest period after five eccentric half-squats at 85% did not improve SJ and CMJ performance variables.

Multi-joint exercises, such as squats and half-squats, are considered effective conditioning stimuli to induce PAPE, leading to improved performance [2,26]. In the present study, a half-squat leading the knee joint lower than 90° was used since it enables greater activation of the lower limb muscle groups [28]. However, our findings demonstrated that the jump tests were not improved after the conditioning stimulus. A previous study has shown that the optimal rest period between the conditioning stimulus and the test varies among individuals and seems to depend on factors such as age, sex, and training background [23]. In the present study, all the participants were male jumper athletes, with eleven of them being long jumpers and the rest of them being high jumpers. Moreover, the participants' age did not vary with a very small standard deviation. Therefore, age, sex, and training background could not be considered responsible for the observed findings.

The sample size and experimental procedure used in the current study allow us to document the effects of the conditioning stimulus on this particular population. Given the above, it is reasonable to affirm that the effects of the specified protocol on jumping performance were recorded. Previous studies focusing on rest periods have demonstrated conflicting results. A rest period of 7–10 min is recommended to optimize power production following a conditioning stimulus. This is despite existing evidence that induction of post-activation potentiation (PAPE) can occur over a spectrum of rest durations, from short to long [23]. However, in the present study, a shorter rest period was used, which mimics the procedure during jumping events at the track and field. Our results showed that the 20 s and

2 min rest periods are quite short to lead to PAPE. The short rest period appears insufficient to induce a potentiation effect, as fatigue seems to prevail over potentiation during this time. However, after 2 min of rest, the emergence of a potentiation effect becomes increasingly evident, suggesting that a longer recovery interval is necessary to obtain the desired performance enhancement. While systematic strength training for jumping events may lead to resistance against muscle fatigue [19,20], the present findings suggest that the 2 min rest period following eccentric half-squats is not sufficient to induce PAPE. Since subsequent power production and performance depend on the balance between potentiation and fatigue, one explanation for the lack of improvement could be the higher level of fatigue compared to the potentiating effect of the conditioning activity. A previous report [38] demonstrated that subjects with a higher percentage of fast-twitch muscle fibers induced a greater PAPE compared to predominantly slow-twitch subjects. However, they also experienced higher fatigue during maximal voluntary contractions [38]. These findings suggest that subjects with a high proportion of fast-twitch fibers may have greater PAPE but may be more prone to experiencing higher fatigue during conditioning when the rest period is short (2 min).

Regarding the lower limbs and jumping ability, it was demonstrated that eccentric half-squats could induce the PAPE effect [29] but not after the specific stimulus and experimental procedure. According to the previous study, eccentric half-squats increased jumping height after the third minute of rest, with most participants maximizing their performance in the sixth minute [29]. Thus, it is likely that the lack of a positive effect of the eccentric half-squat might be due to the short rest period. One possible explanation for the inefficiency of the short rest period is the longer "time under tension" of the knee extensors. It has been previously proposed that the participants tend to decrease their velocity during squats, increasing the time under tension in the muscle and leading to low-frequency fatigue [29,39]. The short rest period likely did not allow the enhancing mechanism to outweigh the inhibitory one, leading to fatigue dominance in this stage.

Regarding the two jumping tasks, our results showed no significant effects after the conditioning stimulus. Our results are in agreement with the previous report, which demonstrated that both eccentric and concentric squats do not affect jumping performance after a short rest period [29]. Even though previous studies [27,40] have shown the efficiency of the high-load stimulus to induce PAPE during the subsequent vertical jump, it seems that our eccentric half-squat stimulus failed to cause any improvements. The specificity of the conditioning stimulus is a determined factor in these contradicting results [27]. Despite the effects of eccentric muscle actions on neural and muscle–tendon systems, they could not improve jumping height when used as a preconditioning stimulus. During SJs, the muscle groups are isometrically activated before the propulsion phase. The differences in muscle activation between the isometric contraction and eccentric action during the jumping test and the conditioning stimulus, respectively, could result in no alteration of the jumping performance. Moreover, the biomechanical differences between the conditioning activity and the jumping test did not affect the SJ jumping height. During the initiation of SJs, participants maintained a 90° knee joint position. In contrast, slightly greater knee flexion was observed during the conditioning activity. Previous research has shown that the depth of the squat can lead to increased activation of the gluteal muscles, while knee extensor activity remains constant [28]. Consequently, the prescribed knee flexion during the SJ test procedure was restrictive, in line with the principle of specificity and the potential for increased gluteal activation. Despite maintaining the same range of motion before and after the stimulus, the conditioning activity did not improve force production. Our data support this notion, as both work and power during the propulsion phase of SJ were stable after the conditioning. The force production during this phase was the same, with the same range of motion leading to the same level of produced work and power after the conditioning activity. Furthermore, the controlled stretching velocity during the eccentric half-squat, compared with the velocity of the eccentric phase during CMJs, could also negatively affect jumping performance. As previously reported, dynamic contractions show joint- and

velocity-specific PAPE effects [23]. It is noteworthy that eccentric velocity alone does not seem to influence PAPE. However, the combination of a low level of eccentric load with a slow eccentric velocity may contribute to the lack of changes observed. It is important to consider that the stretch velocity during eccentric half-squats is lower than that experienced during the eccentric phase of the CMJ. This difference in velocity could potentially account for the lack of discernible effects. Further research is warranted to elucidate the relationship between stretching velocity and PAPE effects.

While this study presents valuable insights into the acute effects of eccentric half-squats on jump performance within a specific population of male athletes specialized in jumping events, certain limitations should be acknowledged. This study's small sample size and exclusive focus on male athletes specializing in jumping events constitute important considerations regarding the generalization of the findings. Furthermore, the decision to exclude athletes from different events based on their distinct training characteristics and potential adaptations is valid. However, it is worth noting that this choice could impact the applicability of the results to a broader athletic context. Another noteworthy limitation involves the variability in strength training backgrounds among participants. Despite the similarities in strength training between the long and high jumpers, individual variations in training adaptation are plausible. Additionally, the absence of gender-based differences in PAPE responses and the lack of female participants raise inquiries about the results' transferability beyond the specific cohort examined. Addressing these limitations in future research endeavors could enrich our understanding of the factors influencing PAPE and its implications for enhancing athletic performance.

## 5. Conclusions

Overall, the findings of this study indicated that the specific conditioning stimulus of eccentric half-squats, followed by a 2 min rest period, did not lead to significant improvements in force capability, the produced work of power, and the overall jumping performance in both SJ and CMJ tests. The results suggest that a single set of eccentric squats with a short rest period may not be effective for enhancing jumping performance in athlete populations like male jumpers. While PAPE remains a valuable concept in sports science, these results highlight the complexity of its application, influenced by factors such as conditioning stimulus specificity, rest period duration, and individual characteristics. Thus, further research is needed to record the possible effects of the PAPE stimulus on the jumping performance and the optimal interaction between the conditioning activity, the rest period, and the athletic population's characteristics.

**Author Contributions:** Conceptualization, E.A. and F.A.; methodology, F.A.; software, G.C.; validation, T.M.K., E.A. and G.C.; formal analysis, E.A. and G.C.; investigation, F.A.; resources, E.A.; data curation, G.C.; writing—original draft preparation, T.M.K.; writing—review and editing, T.M.K., C.P. and N.B.; visualization, G.C.; supervision, F.A.; project administration, F.A.; funding acquisition, F.A. All authors have read and agreed to the published version of the manuscript.

**Funding:** This research received no external funding.

**Institutional Review Board Statement:** This study was conducted in accordance with the Declaration of Helsinki and approved by the Institutional Review Board (or Ethics Committee) of the Department of Physical Education and Sport Science, Aristotle University of Thessaloniki, Serres, Agios Ioannis, Greece (ERC-008/2020 and 23 August 2020).

**Informed Consent Statement:** Informed consent was obtained from all subjects involved in the study.

**Data Availability Statement:** The data presented in this study are available on request from the corresponding author. The data are not publicly available due to privacy.

**Acknowledgments:** We thank the subjects for the time and energy they invested in this study.

**Conflicts of Interest:** The authors declare no conflicts of interest.

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
