# Peer review of "Evaluating the Efficacy of Eccentric Half-Squats for Post-Activation Performance Enhancement in Jump Ability in Male Jumpers"

_applsci, doi:10.3390/app14020749_

Round 1

Reviewer 1 Report

Comments and Suggestions for Authors

The study is interesting, and the manuscript is rather well written. Below you can find some specific comments:

Abstract

Please, include descriptive information about participants (Mean±SD): age, height, body mass; an the level of the jumpers (amateurs, elite…)

Introduction

Line 59. You claim “The high-loaded resistance stimulus typically involves multi-joint free-weight execises with loads exceeding 85%”. What is this information based on? A citation is suggested. In adittion, please add RM: 85% RM.

You analyze and evaluate Work Ecc, Wirk Con, POwer Ecc, Power, Power Con, Lower Leg Stiffness, but you do not refer to them in the introduction. Please complete this section with more information about possible variations in them with strength training.

Material and methods

Please, include information about jumpers levels (amateurs, elite...) and the experience in strength program training (years).

Discussion

Line 214. You claim “Therefore age, sex and training background could not be considered responsible for the observed findings”. You perfectly justify how the training program applied is not enough, but regarding age, would it have worked in younger people? in women? Can the experience and level of the jumpers also influence?

Author Response

Dear reviewers,

We sincerely appreciates your valuable comments provided, and we are confident that their insights will significantly enhance the overall quality of our manuscript.

Reviewer 1

  • The study is interesting, and the manuscript is rather well written. Below you can find some specific comments: Abstract Please, include descriptive information about participants (Mean±SD): age, height, body mass; an the level of the jumpers (amateurs, elite…)

Answer

Information was included in the abstract

The jumping height, peak power, and work were measured and evaluated in twenty male jumpers (age: 21.2 ± 1.7 years, height: 191.1 ± 3.3 cm, body mass: 81.56 ± 7.3 kg) who participated in the national championship last year.

  • Introduction Line 59. You claim “The high-loaded resistance stimulus typically involves multi-joint freeweight execises with loads exceeding 85%”. What is this information based on? A citation is suggested.

Answer

Previous studies suggest that a high loaded resistance stimulus could produce PAPE effect. In most cases high-loaded stimulus represents a >85% RM loading. Moreover, in most cases the dynamic conditioning activity consist of multi-joint actions, such as semi squats, quarter- squats (both concentric and/ or eccentric) for lower limbs  or bench press for the upper limbs.  

Also the citations below have been added

Dobbs et al., 2019; Hincapié et al., 2021; Wilson et al., 2013b

Dobbs, W. C., Olusso, D. V, FEdewa, M. V, & Esco, M. R. (2019). Effect of postactivation potentiation on explosive vertical jump: a systematic review and meta-analysis. Journal of Strength & Conditioning Research, 33(7), 2009–2018.

Hincapié, E. A., Velásquez, A. C. A., Uribe, O. M., García Torres, C. A., & Jaramillo, R. A. (2021). Unilateral and Bilateral Post-Activation Performance Enhancement on Jump Performance and Agility. International Journal of Environmental Research and Public Health, 18(19). https://doi.org/10.3390/ijerph181910154

Wilson, J. M., Duncan, N. M., Marin, P. J., Brown, L. E., Loenneke, J. P., Wilson, S. M. C., Jo, E., Lowery, R. P., & Ugrinowitsch, C. (2013a). Meta-analysis of postactivation potentiation and power: Effects of conditioning activity, volume, gender, rst periods, and training status. Journal of Strength & Conditioning Research, 27(45), 854–859.

 In adittion, please add RM: 85% RM.

Answer

It was added

  • You analyze and evaluate Work Ecc, Wirk Con, POwer Ecc, Power, Power Con, Lower Leg Stiffness, but you do not refer to them in the introduction. Please complete this section with more information about possible variations in them with strength training.

Answer

A paragraph about the importance of the above parameters in the athletic performance was added in the introduction.

The importance of power and work production, coupled with the levels of leg stiffness, during the athletic performance has been previously described (Cormie et al., 2009; Jiménez-Reyes et al., 2014; Maloney & Fletcher, 2021; Morin & Samozino, 2016; Suarez-Arrones et al., 2020). Power production both in eccentric and concentric phase have been described as a determined factor about the final jumping performance (Arabatzi & Kellis, 2012; Jiménez-Reyes et al., 2017).The ability to rapidly produce high levels of force is directly linked to the capacity for work, allowing athletes to execute powerful jumps. The optimal level of leg stiffness ensures the proper transmission of the body from the eccentric to the concentric phase of the stretch shortening cycle, leading to an efficient transfer of the produced work (Maloney & Fletcher, 2021; Walsh et al., 2004). However, despite the importance of these parameters to the jumping performance, the final jumping height seems to be best indicator of an optimal used of the produced force and an efficient transmission of produced power and work. Given the above, the analysis of the kinetic parameters provide important insights for the jumping performance, however the final jumping height seems to be the best indicator of an optimal jump.

  • Material and methods Please, include information about jumpers levels (amateurs, elite...) and the experience in strength program training (years).

Answer

Information were included

All the participants were athletes of jumping events in track and field (12 long jumpers and 8 high jumpers) who participated in the national championship last year. They had been engaged in regular strength training for a minimum of 3 years. Furthermore, these athletes had undergone at least two lower extremity strength development training sessions during the period of peak strength increase in the last 2 years.

  • Discussion Line 214. You claim “Therefore age, sex and training background could not be considered responsible for the observed findings”. You perfectly justify how the training program applied is not enough, but regarding age, would it have worked in younger people? in women? Can the experience and level of the jumpers also influence?

Answer

Unfortunately, the current experimental design did not allow for the investigation of the effects of gender and age on PAPE; however, an alternative experimental approach that specifically addresses these factors could provide valuable insights. The present study and experimental procedures focus exclusively on a specific population, namely jumpers, and assess a potential PAPE. The current experimental design appears unsuitable for investigating potential variations in responses across sexes and/or different age groups. Despite previous studies demonstrating PAPE effects in such groups, our observation that the eccentric stimulus of 85% RM did not induce any PAPE effect in experienced strength training athletes implies that the reported findings might similarly fail to reflect changes in women or younger athletes. Given that the PAPE is related to force capability, and combined with the observed lack of any changes in the jumpers’ performance, it is logical to assume that similar results would be found in women and younger adults. However, this assumption is speculative, and the outcomes depend on several factors, such as conditioning stimulus, rest interval, and training experience.

Reviewer 2 Report

Comments and Suggestions for Authors

The study examines the effects of eccentric half-squats exercise on vertical jumping measures. In general, the study is well written, the study design and the methodology are of acceptable quality. What seems to be missing is a more in-depth explanation of the theoretical background behind post-activation potentiation and a more detailed presentation of the methods and calculations. Why should we expect any effects from an eccentric exercise? a mechanistic approach on muscle physiology would be required here. Further, what was the rationale for the 20s and 2min rest periods?

Abstract

Results description is very short, the most important results with numerical values should be included in the abstract.

Introduction

As already mentioned, muscle physiological functions should be described in relation to PAP including references for eccentric exercise and rest periods. The authors mention that the practical applicability of the high-loaded exercises in inducing PAPE remains unclear (Ln69-71), why? What is more specifically unclear? and how this study aims to contribute to this topic? Please, provide an explanation.

Methods

It is reasonable to think that a slow eccentric exercise (4 sec descending) would not have any effect on vertical jumping, SJ does not have any eccentric phase and CMJ requires a much more explosive movement. Therefore, why didn’t the authors use a faster velocity in squats? Please, add an explanation. Work and leg stiffness are reported in the results, but description of their calculation method is missing. In addition, how was eccentric and concentric phase defined during CMJ. For SJ, how did the authors control for any countermovement.

Sentences in Ln156-160 are redundant and should be moved to the results or removed from the manuscript.

Results

The presentation of the results seems inconsistent. It is not clear why the descriptive results at the three time points are presented only for CMJ, but not for SJ. I see little relevance to present the force-time curves at three time points for SJ and CMJ, since there was not any difference between them. The authors refer to figure 2 for the illustration of maximum power results, but figure 2 shows only force in relation to time and only for one participant. The same is true for leg stiffness during CMJ (Ln188-189).

Eccentric work differed between time points, but there is no post-hoc test for pairwise comparisons.

Please provide and analysis in the discussion for: (1) jump height, which first decreased at post1 time and then increased at post2 time (is this an expected variability? variability due to measurement error?) and (2) eccentric work gradually increased from pre to post2 time points. Is this increase attributed to larger force or to larger distance during countermovement?

Discussion

An in-depth interpretation of the results in relation to the resting period is required. Please consider rephrasing for better clarity: Ln212, Ln217-219, Ln220-222, Ln237 (it is not clear whether the authors refer to the results of the current study or to literature reports), Ln264 (higher glutei muscles…activation?), Ln275-276.

Author Response

Dear reviewers,

We sincerely appreciates your valuable comments provided, and we are confident that their insights will significantly enhance the overall quality of our manuscript.

Reviewer 2

The study examines the effects of eccentric half-squats exercise on vertical jumping measures. In general, the study is well written, the study design and the methodology are of acceptable quality. What seems to be missing is a more in-depth explanation of the theoretical background behind post-activation potentiation and a more detailed presentation of the methods and calculations.

  • Why should we expect any effects from an eccentric exercise? a mechanistic approach on muscle physiology would be required here.

Answer

Despite that eccentric muscle action is an isolated condition compared to usual muscle action, it has been used as a conditioning stimulus in several previous studies (Gołaś et al., 2016; Krzysztofik et al., 2020, 2022). The results of these studies are inconclusive, with some reporting significant positive effects (Beato et al., 2019), some showing non-significant positive effects (Bogdanis et al., 2014), and others showing negative results (Ulrich & Parstorfer, 2017). The present study provides insight into the potential PAPE phenomenon, as our experimental design focuses exclusively on a specific population, namely jumpers. There is currently limited data on the effects of an eccentric stimulus on such a specific population, as seen in track and field jumpers, and this study targets to address this gap in knowledge. A detailed mechanical analysis of the eccentric action is not considered necessary and would not add significantly to the overall content of the manuscript.

  • Further, what was the rationale for the 20s and 2min rest periods?

Answer

The rationale for choosing 20 second and 2 minute rest periods was to record the immediate effects of the eccentric conditioning stimulus on jumping performance. Previous studies have used a rest periods ranging from 0 seconds to 21 minutes. Many propose as the optimal rest length 3-10 minutes. However, a 2-minute rest period could be found in the literature. For information please check (Dobbs et al., 2019; Gouvêa et al., 2013). The choice of the 2 min rest is double fold: firstly, the 2-minute rest period fit very well with the shorter recovery periods often experienced by athletes during competition. Thus, it will provide us data about the use eccentric stimulus as a conditioning stimulus during a competition. Secondly, considering the potential use of eccentric action in the resistance elements of complex/contrast training, a 2-minute rest between exercises was proposed to optimize the training outcomes (Mihalik et al., 2008) (It is a part of Contrast training project). It is possible that the recommended rest period of 2 min to be responsible for the non significant acute effects of this training. Therefore, we thought that the choice of the 2 min res period is acceptable. 

Abstract

  • Results description is very short, the most important results with numerical values should be included in the abstract.

Answer

Information was included

  • Introduction

As already mentioned, muscle physiological functions should be described in relation to PAP including references for eccentric exercise and rest periods. The authors mention that the practical applicability of the high-loaded exercises in inducing PAPE remains unclear (Ln69-71), why? What is more specifically unclear? and how this study aims to contribute to this topic? Please, provide an explanation.

Answer

The present study provides insight into the potential PAPE phenomenon, as our experimental design focuses exclusively on a specific population, namely jumpers. There is currently limited data on the effects of an eccentric stimulus on such a specific population, as seen in track and field jumpers, and this study targets to address this gap in knowledge.

Methods

  • It is reasonable to think that a slow eccentric exercise (4 sec descending) would not have any effect on vertical jumping, SJ does not have any eccentric phase and CMJ requires a much more explosive movement. Therefore, why didn’t the authors use a faster velocity in squats? Please, add an explanation.

Answer

We use the methodology from previous published studies (Koźlenia & Domaradzki, 2023; Krzysztofik et al., 2020, 2022) (citations were included in the manuscript). In most cases a metronome was used to perform an action of about 3 seconds. Our range of motion was a little bit greater than 90⁰. Thus, by the velocity of 30⁰.sec-1, the participants perform an eccentric action lasting more than 3 s. Moreover, the explanation about this choice was that our participants were not familiar with the isolate eccentric action and the choice of a faster velocity might result in inhibitory mechanism as proposed previously by Amiridis et al. (1996). We thought that if the produced force during conditioning is a determinate component of the following adaptation, it would be much safer to use a slower eccentric velocity.  

  • Work and leg stiffness are reported in the results, but description of their calculation method is missing.

Answer

The description of their calculation were included in the manuscript

  • In addition, how was eccentric and concentric phase defined during CMJ. For SJ, how did the authors control for any countermovement.

Answer

Both pure concentric phase of SJ and the define of two phases of CMJ was controlled by two ways: by the curve force-time and by the high speed video recording.

 For the SJ, there was no reduction in the initial values of the vertical force indicating that only concentric action took place. For the CMJ, the eccentric phase was defined as the return of the vertical force to its initial level in the upright position. Moreover, a video of each jump was recording to ensure the proper technique of the participants.

  • Sentences in Ln156-160 are redundant and should be moved to the results or removed from the manuscript.

Answer

They were removed

  • Results

The presentation of the results seems inconsistent. It is not clear why the descriptive results at the three time points are presented only for CMJ, but not for SJ. I see little relevance to present the force-time curves at three time points for SJ and CMJ, since there was not any difference between them. The authors refer to figure 2 for the illustration of maximum power results, but figure 2 shows only force in relation to time and only for one participant. The same is true for leg stiffness during CMJ (Ln188-189).

Answer

We apologize for reference about Figure 2. It was a mistake by oversight. We delete the references. Moreover we add a new table (referred as Table 1) to present the descriptive data for the SJ.

  • Eccentric work differed between time points, but there is no post-hoc test for pairwise comparisons.

Answer

We apologize for the omission. The sentence about pos-hoc test was added.

Post-hoc pairwise comparisons using Bonferroni revealed a significant mean difference between pre (M=89.7J) and post1 (M=100.1J; p = 0.027) and between pre and post2 (M=104.9; p = 0.008) while no significant mean difference was found between post1 and post2 (p = 0.562) for eccentric work.

  • Please provide and analysis in the discussion for: (1) jump height, which first decreased at post1 time and then increased at post2 time (is this an expected variability? variability due to measurement error?)

Regarding the jumping height, we thought that the possible fatigue immediately after the stimulus might be a good explanation about the decrease. The 20 s rest period might be two short, as showed by previous studies, to overcome the produced fatigue from the stimulus. After 2 min rest, the potentiating effect becomes prominent.  

and (2) eccentric work gradually increased from pre to post2 time points. Is this increase attributed to larger force or to larger distance during countermovement?

The increase in produce force attribute to higher produced force during eccentric phase, since the range of motion during the CMJ, the range of motion of the knee was predetermined (90⁰).

Discussion

  • An in-depth interpretation of the results in relation to the resting period is required.

The paragraph regarding the resting period has been changed

The sample size and experimental procedure used in the current study allow us to document the effects of the conditioning stimulus on this particular population. Given the above, it is reasonable to affirm that the effects of the specified protocol on jumping performance were recordedPrevious studies focused on rest periods have demonstrated conflicting results. To optimise power production following a conditioning stimulus, a rest period of 7-10 minutes is recommended. This is despite existing evidence that induction of post-activation potentiation (PAPE) can occur over a spectrum of rest durations, from short to long (Wilson et al., 2013a).  However, in the present study, a shorter rest period was used, which mimics the procedure during jumping events at the track and field. Our results showed that the 20s and 2min rest periods is quite short to lead to PAPE. The short rest period appears to be not sufficient to induce a potentiation effect, as fatigue seems to prevail over potentiation during this time. However, after 2 min of rest, the emergence of a potentiation effect becomes increasingly evident, suggesting that a longer recovery interval is necessary to obtain the desired performance enhancement.While systematic strength training for jumping events may lead to resistance against muscle fatigue [19,20], the present findings suggest that the 2-minute rest period following eccentric half-squat is not sufficient to induce PAPE. Since subsequent power production and performance depend on the balance between potentiation and fatigue, one explanation for the lack of improvement could be the higher level of fatigue compared to the potentiating effect of the conditioning activity. Previous report (Hamada et al., 2000)demonstrated that subjects with a higher percentage of fast twitch muscle fibers induced a greater PAPE compared to predominantly slow twitch subjects. However, they also experienced higher fatigue during maximal voluntary contractions (Hamada et al., 2000). These findings suggest that subjects with a high proportion of fast twitch fibers may have greater PAPE, but may be more prone to experience higher fatigue during conditioning when the rest period is short (2min). 

 Please consider rephrasing for better clarity: Ln212, Ln217-219, Ln220-222, Ln237 (it is not clear whether the authors refer to the results of the current study or to literature reports), Ln264 (higher glutei muscles…activation?), Ln275-276.

All the sentences were rephrased.

Reviewer 3 Report

Comments and Suggestions for Authors

This is such an interesting study and I believe that there is a merit for this journal. Authors did such a nice  systematic review and conducted this study. I believe that reliability and validity of tests and measurements should be reported for each specific test in more detail. In addition, authors should add more up to date literature review (2020 and above) and practical suggestions and future recommendations should be added. Thank you and best regards.

Author Response

Dear reviewers,

We sincerely appreciates your valuable comments provided, and we are confident that their insights will significantly enhance the overall quality of our manuscript.

Reviewer 3

This is such an interesting study and I believe that there is a merit for this journal. Authors did such a nice  systematic review and conducted this study. I believe that reliability and validity of tests and measurements should be reported for each specific test in more detail.

Answer

They were added in the manuscript at the result section

The Intraclass Correlation Coefficients (ICCs) for the Squat Jump (SJ) and Countermovement Jump (CMJ) measurements yielded values of 0.92 and 0.89, respectively, indicating high reliability and consistency in these assessment techniques.

 In addition, authors should add more up to date literature review (2020 and above) and practical suggestions and future recommendations should be added. Thank you and best regards.

Round 2

Reviewer 2 Report

Comments and Suggestions for Authors

I would like to thank the authors for their revision addressing most of the comments during first round review. I still miss a convincing explanation of why to use eccentric muscle contractions to enhance performance in vertical jump, however, I leave this question to the editors.

Some minor comments:

Ln76: please provide explanation of abbreviations when first appear (e.g. RM – repetition maximum)

Ln173: symbol for describing difference in vertical displacement is not consistent (ΔL vs. Δl). Please unify upper and lower case. Difference in vertical displacement was between standing height and at the lowest position?

Ln174-175: the description…in the middle of the ground contact phase is not accurate. Nor for displacement, neither for time the lowest position during countermovement is in the middle. The lowest position is when vertical velocity is equal with zero.

Ln176: Similarly, the authors mention that Fpeak occurred at the lowest position. This is quite common in vertical jumping, however, not always true by default. Fpeak may occur also before or shortly after the lowest position. This needs to be more precisely described.

In table 1 font type is not consistent.

Reference nr 17 has a different format than the rest references, please check.

Please consider the use of oxford comma throughout the entire manuscript.

Comments on the Quality of English Language

The use of oxford comma throughout the entire manuscript may be considered.

Author Response

I would like to thank the authors for their revision addressing most of the comments during first round review. I still miss a convincing explanation of why to use eccentric muscle contractions to enhance performance in vertical jump, however, I leave this question to the editors.

Some minor comments:

Ln76: please provide explanation of abbreviations when first appear (e.g. RM – repetition maximum)

The explanation of abbreviation was included

Ln173: symbol for describing difference in vertical displacement is not consistent (ΔL vs. Δl). Please unify upper and lower case. Difference in vertical displacement was between standing height and at the lowest position?

We use Dl in all cases. Also, the sentence was modified “...force to Δl at the instant in the middle of the ground contact phase when the COM reached its lowest point from the standing position, and the leg spring was maximally compressed.”

Ln174-175: the description…in the middle of the ground contact phase is not accurate. Nor for displacement, neither for time the lowest position during countermovement is in the middle. The lowest position is when vertical velocity is equal with zero.

The sentence was modified as

"Leg stiffness was calculated as the ratio of the ground reaction force to Δl at the moment when the vertical velocity equaled zero, while the center of mass (COM) was at its lowest point from the standing position and the leg spring was maximally compressed."

n176: Similarly, the authors mention that Fpeak occurred at the lowest position. This is quite common in vertical jumping, however, not always true by default. Fpeak may occur also before or shortly after the lowest position. This needs to be more precisely described.

The sentence was modified

“The peak vertical ground reaction force (Fpeak) was obtained from the force- time curve usually while the leg was maximally compressed (Fpeak may also displayed either slighty before of immediately after the lower positions).”

In table 1 font type is not consistent.

We change the fond.

Reference nr 17 has a different format than the rest references, please check.

We change the authors’ names from capitals to lower case.

Please consider the use of oxford comma throughout the entire manuscript.

We used oxford comma throughout the entire manuscript